# Promising Antimicrobial Action of Sustained Released Curcumin-Loaded Silica Nanoparticles against Clinically Isolated *Porphyromonas gingivalis*

**DOI:** 10.3390/diseases11010048

**Published:** 2023-03-08

**Authors:** Adileh Shirmohammadi, Solmaz Maleki Dizaj, Simin Sharifi, Shirin Fattahi, Ramin Negahdari, Mohammad Ali Ghavimi, Mohammad Yousef Memar

**Affiliations:** 1Department of Periodontics, Faculty of Dentistry, Tabriz University of Medical Sciences, Tabriz 5166, Iran; 2Dental and Periodontal Research Center, Tabriz University of Medical Sciences, Tabriz 5166, Iran; 3Department of Dental Biomaterials, Faculty of Dentistry, Tabriz University of Medical Sciences, Tabriz 5166, Iran; 4Department of Oral and Maxillofacial Pathology, Faculty of Dentistry, Tabriz University of Medical Sciences, Tabriz 5166, Iran; 5Department of Prosthodontics, Faculty of Dentistry, Tabriz University of Medical Science, Tabriz 5166, Iran; 6Department of Oral and Maxillofacial Surgery, Faculty of Dentistry, Tabriz University of Medical Sciences, Tabriz 5166, Iran; 7Infectious and Tropical Diseases Research Center, Tabriz University of Medical Sciences, Tabriz 5166, Iran

**Keywords:** antimicrobial action, sustained release, curcumin-silica nanoparticles, periodontal diseases, *Porphyromonas gingivalis*

## Abstract

Background. *Porphyromonas gingivalis* (*P. gingivalis*) has always been one of the leading causes of periodontal disease, and antibiotics are commonly used to control it. Numerous side effects of synthetic drugs, as well as the spread of drug resistance, have led to a tendency toward using natural antimicrobials, such as curcumin. The present study aimed to prepare and physicochemically characterize curcumin-loaded silica nanoparticles and to detect their antimicrobial effects on *P. gingivalis*. Methods. Curcumin-loaded silica nanoparticles were prepared using the chemical precipitation method and then were characterized using conventional methods (properties such as the particle size, drug loading percentage, and release pattern). *P. gingivalis* was isolated from one patient with chronic periodontal diseases. The patient’s gingival crevice fluid was sampled using sterile filter paper and was transferred to the microbiology laboratory in less than 30 min. The disk diffusion method was used to determine the sensitivity of clinically isolated *P. gingivalis* to curcumin-loaded silica nanoparticles. SPSS software, version 20, was used to compare the data between groups with a *p* value of <0.05 as the level of significance. Then, one-way ANOVA testing was utilized to compare the groups. Results. The curcumin-loaded silica nanoparticles showed a nanometric size and a drug loading percentage of 68% for curcumin. The nanoparticles had a mesoporous structure and rod-shaped morphology. They showed a relatively rapid release pattern in the first 5 days. The release of the drug from the nanoparticles continued slowly until the 45th day. The results of *in vitro* antimicrobial tests showed that *P. gingivalis* was sensitive to the curcumin-loaded silica nanoparticles at concentrations of 50, 25, 12.5, and 6.25 µg/mL. One-way ANOVA showed that there was a significant difference between the mean growth inhibition zone, and the concentration of 50 µg/mL showed the highest inhibition zone (*p* ≤ 0.05). Conclusion. Based on the obtained results, it can be concluded that the local nanocurcumin application for periodontal disease and implant-related infections can be considered a promising method for the near future in dentistry.

## 1. Introduction

The main causes of periodontal diseases are inflammation and infection of the gums and bone surrounding the teeth. In the early stage of periodontal disease, which is called gingivitis, the gums become swollen and red and may bleed. In more advanced stages, called periodontitis, the gums can separate from the teeth, the bone can be lost, and the teeth can become loose or even fall out. The two biggest threats to dental health are tooth decay and periodontal disease [1,2]. These diseases affect the tissues supporting and protecting the teeth and can deteriorate the alveolar bone and periodontal ligament. Their frequency and severity vary greatly between communities; nonetheless, it is expected that 15 to 20 percent of adults are infected with the more severe types of illness, while 35 to 60 percent of the population is afflicted with less severe conditions [3,4]. The most important goal of treating periodontitis is to completely clean the pockets around the teeth and inhibition of surrounding bone damage. If the periodontitis has not progressed much, treatment may include less invasive and nonsurgical methods, including root planing (smoothing surfaces of the root, preventing further accumulation of bacteria and plaque), scaling (eliminating bacteria and plaque from surfaces of the tooth), and the oral or topical application of antibiotics [5]. Patients with advanced periodontitis may need dental surgery for treatment, including pocket reduction surgery (flap surgery), bone grafting, soft tissue grafting, and the regeneration of guided tissue [6].

Bacterial infections in dentistry may induce implant-associated issues, resulting in tissue and organ function loss or even implant failure. In dentistry, bacterial infections contribute to the development of caries and periodontitis, which are two of the most prevalent bacterial infections in humans [7]. Numerus bacterial strains are involved in periodontal disease development, such as *Aggrigatibacter actinomycetemcomitans*, *Capnocytophaga* species and *Eikenella corrodens*, *P. gingivalis*, *Tannerella forsythia*, *Treponema denticola*, *Prevotella intermedia*, *Actinomyces* species, and *Fusobacterium nucleatum* [8]. *Porphyromonas gingivalis* (*P. gingivalis*) has always been one of the leading causes of periodontal disease, and antibiotics are commonly used to control it. Numerous systems releasing antibacterial and remineralizing substances, such as fluoride (F), calcium (Ca^2+^) as well as phosphate (PO_4_^3–^), or silver (Ag^+^) ions, have been described for effective avoidance or treatment of biofilm infections [9,10]. Chlorhexidine (CHX) is frequently used because of its excellent antibacterial efficacy against Gram-negative and Gram-positive microorganisms, fungi, and viruses [11]. Furthermore, the propensity to produce resistance is minor [12]. Preservatives are also included in disinfection products and oral rinses [13]. Moreover, by attaching to the enamel and pellicle, CHX suppresses the production of bacterial biofilms. The first stage in creating biofilms, i.e., the aggregation of bacterial cells on these surfaces, is inhibited [13,14]. CHX has a significant substantivity that indicates a long-term interaction with particular substrates, such as tooth surfaces or mucosa inside the oral cavity [14]. This elevated CHX is the gold standard for dentistry microbial infection prevention and treatment [14]. However, tooth staining is one drawback of CHX, which limits its long-term usage. There are also other side effects, including tongue and mucosal surface staining, changes of taste, desquamation of the mucosa, expansion of the parotid and enlarged calculus deposition supragingivally [15]. In addition, the spread of drug resistance has led to a tendency toward using new natural antimicrobials [16].

The active ingredients in plants are widely used in the treatment of various diseases [17,18]. Curcumin is a substance produced from the rhizomes of the *Curcuma longa* plant, and it is commonly utilized in culinary applications [17,18]. A wide variety of publications have reported on curcumin’s anti-inflammatory, wound-healing, antimicrobial, and anti-neoplastic properties, utilized in *in vitro* and *in vivo* strategies for conditions ranging from diabetes to neurological disturbances, in cancer, in autoimmune disorders, and in chronic inflammatory conditions, including Crohn’s disease, rheumatoid arthritis, and periodontal disease [19]. Curcumin’s anti-inflammatory properties have been found to diminish immune cell response to periodontal disease-associated bacterial antigens and to restrict periodontal tissue destruction in *in vitro* and *in vivo* studies [19,20]. Nevertheless, since most of this *in vivo* research has utilized a systemic manner of administration, curcumin’s poor pharmacodynamic properties, including hydrophobicity, low gastrointestinal absorption rate, and very short plasma half-life, may have skewed their results [21].

New designs based on nanotechnology have been discovered to improve the bioavailability of curcumin and reduce its cytotoxicity [22]. Today, nanotechnology has become important in various medical fields, such as drug delivery [23]. Nanoporous silica materials have been extensively studied [24,25,26] since their initial deployment as a drug delivery platform in 2001 [27] or as implant surface coatings [28,29]. Nanoporous silica has a variety of qualities that make it an attractive option for a controlled-release system. It has a large surface area, huge pore volumes, and variable pore sizes with contracted pore size distributions, allowing for significant cargo loading. On the other hand, uncontrolled antimicrobial chemical leaching from release mechanisms has disadvantages. Although burst release could benefit the treatment of acute infections, and it is much more efficient than protracted delivery, it is essential for controlled release systems that can stay quiescent for lengthy periods yet distribute cargo when triggered. As a result, the medicine remains in the pores and could be removed when required. Due to the antimicrobial properties of curcumin and the useful characteristics of porous silica nanoparticles as a sustained-release carrier, the present study was conducted with the aim of preparing and physicochemically identifying curcumin-loaded silica nanoparticles and evaluating their antimicrobial effect on *P. gingivalis*.

## 2. Material and Methods

### 2.1. Preparation of Mesoporous Silica Nanoparticles Containing Curcumin

Fifteen milligrams of powder of silica nanoparticles (Nano Sadra Company, Isfahan, Iran) and 0.75 mg of curcumin powder (Sigma Aldrich, Burlington, MA, USA) were added to 10 mL of cyclohexane. The prepared suspension was sonicated, stirred overnight, and washed with cyclohexane, and the silica particles containing curcumin were vacuum dried [30]. The nanoparticles were stored at −18 °C for further investigations.

### 2.2. Sampling of P. gingivalis

To attain clinically isolated *P. gingivalis*, one patient with chronic periodontal disease was selected from the patients referred to the Department of Periodontics, Faculty of Dentistry, Tabriz University of Medical Sciences, Tabriz, Iran. With sterile gauze, the surface of the tooth was cleaned, and the gingival crevice fluid was then sampled using sterile filter paper and placed in a thioglycollate broth media. The samples were moved to the microbiology laboratory in less than 30 minutes and stored at −20 °C until assayed.

### 2.3. Cultivation of P. gingivalis

The isolated sample from the mentioned patient was vortexed for 30 s. Selective medium for *P. gingivalis* containing Columbia agar base supplemented with vitamin K1, 5% defibrillated sheep blood, hemin, colistin sulfate, bacitracin, and nalidixic acid was used [31]. Then, the plates were incubated under 80% N_2_, 10% CO_2_, 10% H_2_ and 0% O_2_ in anaerobic conditions provided by the Anoxomat system (MART microbiology B.V., Drachten, The Netherlands). The growth of bacterial colonies was examined at 48, 72, and 96 h. The trypsin reagent test was used to confirm the presence of *P. gingivalis* on the plates. Gingipain, which is produced by *P. gingivalis,* is a trypsin-like enzyme. The aerotolerance test and biochemical and microbiological assays (such as colony morphology, special potency disks, pigment production, fluorescent under UV light, catalase test, indole, and trypsin-like peptidase activity assay) were used to identify *P. gingivalis* isolates [31].

### 2.4. Characterization of the Nanoparticles

#### 2.4.1. The Particle Size of Nanoparticles

The prepared nanoparticles were characterized using a dynamic light scattering (DLS) device (DLS, Malvern, Cambridge, UK) for size determination. The suspension of the nanoparticles was prepared in distilled water and poured into the device. An argon laser beam at 633 nm and a scattering angle of 90° at 25 °C were used for DLS device settings. DLS is an instrument for measuring the hydrodynamic size of molecules and submicron and nanoparticles. This test was performed three times.

#### 2.4.2. Morphology and the Cytotoxicity Investigation

Transmission electron microscopy (TEM) is a powerful tool to investigate the interaction of nanoparticles, their structure, and their morphology. A transmission electron microscope (TEM-2100F; JEOL, Tokyo, Japan) was used to investigate the mesoporous structure of the silica nanoparticles. For this analysis, the samples were prepared by dropping a solution of nanoparticles in deionized water on a carbon-coated copper TEM grid, followed by imaging. Size histograms for free silica nanoparticles and curcumin-loaded silica, based on TEM analysis, were also reported.

Cell viability examination was used to define the cytotoxicity of the prepared nanoparticles against dental pulp stem cells. The cells were obtained from the cell bank of Shahid Beheshti University (Tehran, Iran). Then, the nanoparticles as disks were placed in the bottoms of the wells. The cells were cultured in a single layer in DMEM including serum and antibiotics. After 72 h, the washing, incubating (for 4 h at 37 °C), and adding of MTT solution (2 mg/mL PBS) were performed. As a next step, the above solution was removed and, 200 mL of DMSO and 25 mL of Sorenson glycine buffer were added to each well. The absorbance was read at 540 nm, and the percentage of living cells was evaluated. Cells grown without any material were considered as control group.

#### 2.4.3. Determination of Curcumin Loading Inside the Nanoparticles

One of the key parameters for drug-loaded nanoparticles is drug loading percentage, which is defined as the mass ratio of drug to drug-loaded nanoparticles. To determine the amount of curcumin loaded on silica nanoparticles, 10 mg of the prepared nanoparticles were dissolved in 20 mL of dimethyl sulfoxide. One milliliter of the dissolved nanoparticle solution was poured into a special tube of an ultraviolet spectrophotometer, and Lambda Max was adjusted to 350 nm for curcumin. This test was performed three times.

#### 2.4.4. Evaluation of Release Pattern

Drug release denotes the procedure in which drug solutes migrate from the initial position in the carrier system to the carrier’s outer surface and then to the release medium. To determine the pattern of drug release from curcumin-loaded silica nanoparticles, phosphate buffer (300 mL) was poured into 3 beakers. An amount of 5 mg of the prepared nanoparticles was poured into the beaker. The pH of the liquid was adjusted to 7.4, and the temperature was set to 37 °C. The stirrer was set to 100 rpm. Indeed, these parameters had to be established based on the body’s condition for a dissolution test of a drug (pH of 7.4, temperature of 37 °C, and stirring rate of 100 rpm). Samples were taken from the beaker every day (1 mL), and the absorbance was noted using a UV spectrophotometer for curcumin at 350 nm. The sample taken from the beakers was replaced with 1 mL of a new buffer medium to keep the concentration in balance. The amount of UV absorption was then changed to concentration. Subsequently, the cumulative release percentage was designed against the time (day) for the release study. The calculation method for the percentage of cumulative release (%) was according to the following equation:Cumulative percentage release (%) = Volume of sample withdrawn (mL)/The volume of release media (v) × P (t − 1) + Pt
where Pt is the percentage release at time t.

#### 2.4.5. The Antimicrobial Action of Nanoparticles

The original method for determining susceptibility to antimicrobials was based on broth dilution methods. In this study, the disk diffusion method as a routine laboratory test was utilized to investigate the antibacterial effects of silica nanoparticles loaded with curcumin. This method identifies the action of bacteria on an antimicrobial material by creating a gradient of concentration around a disk. The bacterial isolate used in this study was isolated from a patient with chronic periodontal disease. First, a bacterial suspension of 0.5 McFarland was prepared, and then, using a sterile cotton swab, a uniform grass culture was grown on the surface of Brucella agar enriched with dried sheep blood (5%), vitamin K1 (1 μg/mL), and hemin (5 μg in mL). To prepare discs containing nanoparticles, sterile blank disks were immersed in concentrations of 3.12, 6.25, 12.5, 25, and 50 μg/mL nanoparticle suspensions, and then the disks were placed on the agar surface. A blank disk was used as a negative control, and metronidazole antibiotic disks (5 μg/mL) were used as a positive control. After incubating the plates at 37 °C for 42 h, the growth inhibition zones were measured. With this method, the halos of non-growth around the discs were measured from the back of the plate with a ruler based on millimeters.

In the next step, Brucella broth supplemented with hemin (5 µg/mL), vitamin K1 (1 µg/mL), and lysed horse blood (5%) in the presence of a serial concentration of nanoparticles (50, 25, 12.5, and 6.25 µg/mL concentrations) was applied to obtain the MICs of the nanoparticles against *P. gingivalis*. The wells were incubated for 48 h at 35 °C and then observed for microbial growth turbidity. The positive control was metronidazole antibiotic, and water was considered as a negative control.

## 3. Statistical Analysis

The results are stated as descriptive indices. The Shapiro–Wilk test was applied to test the normality of the units. The, we used SPSS software, version 20 (IBM Company, Armonk, NY, USA), to compare the data between groups with a *p* value of <0.05 as the significance level. One-way ANOVA and Tukey’s post hoc test were utilized to compare the groups. The flow chart of the study process is shown in Figure 1.

## 4. Results and Discussion

The low bioavailability of curcumin is the most important concern for its clinical use. Additionally, little information is available about its safety at higher doses. Today, to reduce its toxicity and improve the bioavailability of curcumin, new designs based on its nanoformulation have been discovered [17,18]. Evaluating the physicochemical properties of nanoparticles is necessary to ensure their suitability for various uses. The interactions of nanoparticles *in vitro* and *in vivo* are related to their physicochemical properties [32]. Reducing the size of nanoparticles increases their surface area, the interaction of these nanoparticles with the environment increases, and their ways of crossing body barriers and entering cells will be different [33,34].

The average particle size of drug-free silica nanoparticles is shown in Figure 2a, and that for curcumin-loaded silica nanoparticles is shown in Figure 2b. The results showed that both types of nanoparticles had nanometric sizes. For drug-free silica nanoparticles the mean particles size was 90 ± 1.02 nm, while curcumin-loaded silica nanoparticles had a mean particle size of 110 ± 1.23 nm. Figure 3a shows the morphology of the drug-free silica nanoparticles, and the morphology of curcumin-loaded silica nanoparticles has shown in Figure 3b. The size histograms for free silica nanoparticles and curcumin-loaded silica, based on TEM analysis, are shown also in the Figure 3c and d, respectively. Our outcomes showed that the nanoparticle sizes differed in DLS analysis compared to TEM analysis. This difference may be owing to the hydrating of the outer layer of the nanoparticles in the DLS technique. In addition, the aggregation of nanoparticles and the non-spherical shape of nanoparticles could be the cause of this difference [35].

Nanoparticles exert their antimicrobial effects on bacteria by several mechanisms that depend on the size of the nanoparticles and the type of bacteria. The dose of nanoparticles and their physicochemical properties (shape, size and surface properties) are very important to their antimicrobial effects [36]. The size of nanoparticles is important to their antibacterial effect, so smaller nanoparticles, by binding to the surface of bacteria with high affinity, can disrupt the function of the cell membrane of bacteria compared to larger nanoparticles [37]. The interaction of nanoparticles with the bacterial membrane causes local pores in the membrane. Additionally the entry of nanoparticles into bacterial cells causes damage to DNA and proteins (especially sulfur-rich proteins). In this way, nanoparticles can disrupt the function of bacteria. Nanocarriers containing antibacterial agents can also combine their structure with the bacterial cell wall and introduce their medicinal substances into the cytoplasm [38].

TEM pictures proved the mesoporous building and the rod-shaped morphology of the prepared nanoparticles. The filled pores of mesoporous silica can also be detected by TEM imaging of drug-loaded mesoporous silica nanoparticles that show the loading of curcumin into the silica nanoparticles. Rod-shaped nanoparticles may display a longer circulation time and a slight uptake by the RES in the body compared with spherical particles [39,40]. A recent *in vivo* study also showed that rod-type nanoparticles exhibit a high capacity to overcome uptake through RES and show a longer presence in the blood compared with spherical nanoparticles [41].

The percentage of cytotoxicity (cell viability) of the prepared nanoparticles on dental pulp stem cells is shown in Figure 4. There was no significant reduction in the viability of the cells exposed to the nanoparticles compared to the control group (cells grown without any material). Therefore, the prepared nanoparticles were non-cytotoxic against dental pulp stem cells (Figure 4).

The loading results showed that the loading percentage of curcumin in silica nanoparticles was 68% ± 1.02. Currently, most nanoparticle systems have relatively low drug loading, and increasing the increase drug loading capacity remains a challenge. The reason for the high drug-loading percentage of our nanoparticles was their mesoporous structure.

The prepared nanoparticles displayed a relatively fast release pattern in the first 5 days (Figure 5). The release of curcumin from silica nanoparticles continued slowly until day 45. The burst release of curcumin from the prepared nanoparticles could eradicate acute infections, and the controlled sustained release could provide the drug content for long periods. As a result, the drug remained in the pores and could be removed when required [42]. It seems that the pattern of rapid drug release from nanoparticles in the first days is related to drugs adsorbed to the surface of nanoparticles that are not inside the cavities and have a weak interaction with the outer surface of the cavities. Curcumin molecules inside the cavities that had electrostatic interactions with the nanoparticle cavity wall caused slow and continuous release on days 6 to 45. The slow-release pattern of drugs is very critical in the clinical application of drugs [43]. Memar et al. achieved similar results for meropenem-loaded silica nanoparticles [44]. They showed that, in the first two days, about 40 percent of meropenem was released from silica nanoparticles, and then slow release was sustained until the 30th day.

With a conventional drug-delivery method, the drug concentration in the blood remains within a relatively large range for a short period of time, which can fall short of the lowest effective dose or exceed the maximum tolerated dose. As a result, frequent doses are necessary, which will be associated with side effects. Using the appropriate nanocarrier, the blood concentration of the drug at the site of infection can be maintained at the required effective concentration for a long time and, as a result, reduce the frequency of consumption, produce good stability, reduce patient pain, and improve patient compliance. The drug loaded in the nanocarrier has a much more prominent inhibitory effect on cell growth with long-term drug release compared to the free drug at the same concentration [45].

### Antimicrobial Action

The results of microbial tests showed that *P. gingivalis* is sensitive to the silica nanoparticles loaded with curcumin at concentrations of 50, 25, 12.5, and 6.25 μg/mL. The mean growth inhibition zones of curcumin-loaded silica nanoparticles concentrations and control antibiotic (metronidazole) are shown in Table 1 and Figure 6.

Based on the MIC test, the nanoparticles showed inhibitory effects against *P. gingivalis* at 6.25 µL/mL. In addition, based on our previous study, free silica nanoparticles did not have any significant antibacterial effects [46].

One-way ANOVA (between curcumin groups) revealed that there is a significant relation in the concentration of curcumin-loaded silica nanoparticles with the size of the growth inhibition, zone and the highest inhibition zone was displayed in the concentration of 50 µg/mL (*p* ≤ 0.05). Tukey’s post hoc test showed that there was a significant difference between the antimicrobial effects of all concentrations of curcumin-loaded silica nanoparticles (*p* ≤ 0.05). Thus, the nanoparticles had dose-dependent antimicrobial effects.

Other studies used *P. gingivalis* (ATCC33277). In a study, Shahzad et al. reported that the growth inhibition of *P. gingivalis* (ATCC33277) was effected by curcumin at a concentration of 7.8 μg/mL [47]. Additionally, Mandroli and Bhat showed that the MIC of curcumin against *P. gingivalis* (ATCC33277) was 125 μg/mL [48], while Izui et al. showed that the prevention of bacterial growth occurred with curcumin at a concentration of 20 μg/mL [49]. In another recent study, the sensitivity of *P. gingivalis* (ATCC33277) to curcumin was shown in a concentration of 100 μg/mL [50]. The main reason for the difference between the results of our study and the results of other studies may be that they investigated the effects of free curcumin on laboratory strains, while in our study, the effects of sustained-release nanoparticles containing curcumin on clinically isolated *P. gingivalis* were investigated.

In our previous study, the prevalence of *P. gingivalis* isolated from the gingival crevicular fluid (GCF) of 15 Iranian patients with implant failure was investigated. The results showed that, out of 15 patients, eight (53.33%) were positive for the presence of *P. gingivalis*. The antimicrobial action of curcumin nanocrystals was also investigated against *P. gingivalis* isolated from patients with implant failure, and the results showed that curcumin nanocrystals had an MBC of 12.5 µg/mL and a MIC of 6.25 µg/mL. Additionally curcumin nanocrystals showed the highest inhibition zone at the concentration of 50 µg/mL (*p* = 0.0003) [51].

A study showed that curcumin prevented bacterial strains by damaging the membrane of bacteria [52]. Curcumin can inhibit the proliferation of bacteria by perturbation of FtsZ assembly. Some studies have shown that curcumin deactivates bacteria by stimulating ROS generation [53,54].

Kumbar and coworkers explained the effects of curcumin on the biofilm formation and virulence factor gene expression of *P. gingivalis* using gene expression studies. They showed that the MBC and MIC of curcumin for both clinical strains and ATCC of *P. gingivalis* were 125 and 62.5 µg/mL, respectively. Curcumin inhibited attachment and biofilm formation of bacteria in a dose-dependent way. Additionally, curcumin decreased the virulence of *P. gingivalis* by decreasing the expression of proteinases (rgpA, rgpB, and kgp) and adhesions (fimA, hagA, and hagB) as the main genes of virulence factors. Curcumin has presented anti-biofilm and antibacterial effects against *P. gingivalis*. Furthermore, due to the pleiotropic actions of curcumin, it can be an inexpensive and readily available therapeutic agent in the treatment of periodontal disease [55].

Chen and coworkers investigated the anti-inflammatory effects and the mechanism of action of curcumin in macrophages stimulated by *P. gingivalis* lipopolysaccharide (LPS). They reported that curcumin prevented the expression of IL-1β and TNF-α genes and protein synthesis in RAW264.7 cells that were stimulated with LPS of *P. gingivalis*. In RAW264.7 cells, LPS of *P. gingivalis* stimulated NF-ĸB-dependent transcription, which was downregulated by pretreatment with curcumin [56].

## 5. The Strengths and Limitations

The results of this investigation showed that curcumin-loaded silica nanoparticles had suitable antibacterial actions against *P. gingivalis*. This finding could be very useful in overcoming bacterial resistance. In addition, the concentrations obtained in this study were lower compared to those obtained previous research works, advancing the hope of preparing optimal formulations based on these nanoparticles.

The main limitation of this study was its use of a single isolate of *P. gingivalis***.** A single isolate is not enough to draw conclusions regarding MIC values and accurately compare them to other studies. In addition, the possibility of human error in the sampling of bacteria, nanoparticle aggregation, and microbial contaminations with other bacterial strains can be considered other limitations.

There are also other types of bacteria that act as periodontal pathogens, such as *Fusobacterium nu-cleatum*, *Prevotella Intermedia*, *Aggregatibacter* and *Actinomicetencomitans*. Curcumin-loaded silica nanoparticles should also be examined against these bacteria in future studies.

This report was an *in vitro* study. Any possible toxicity of these nanoparticles should be tested in future studies before any animal or clinical trials. Moreover, the antimicrobial and antibiofilm mechanisms for them should be investigated to confirm their exact function.

## 6. Suggestions and Future Perspective

It is suggested to investigate the effects of curcumin-loaded silica nanoparticles on *P. gingivalis*-related infections *in vivo* and then clinically. Additionally silica nanoparticles co-loaded with curcumin and other antibacterial agents can be prepared, and their antibacterial effects can be investigated *in vitro*, *in vivo*, and clinically. A limited number of clinical isolates of *P. gingivalis* were analyzed in this study, and they can be used in future studies to investigate the effects of curcumin-loaded silica nanoparticles on a greater number of bacteria.

Nanoformulations of plant substances or phytochemicals can replace chemical antibacterial drugs in the future. This replacement can be a solution to reduce the use of antibiotics, which will reduce not only microbial resistance but also the toxicity and side effects caused by antibiotics.

## 7. Conclusions

This study showed that *P. gingivalis* clinically isolated from the gingival crevice fluid of a patient with chronic periodontal diseases is highly sensitive to curcumin-loaded silica nanoparticles at a low concentration. In addition, the two-stage release profile of the prepared nanoparticles can provide both the burst release and the controlled sustained release of curcumin, which can be used to eradicate acute infections at first and then provide the drug content for a long time. It can be concluded that local nanocurcumin application for periodontal disease and implant-related infections can be considered as a promising method for the near future in dentistry.

## Figures and Tables

**Figure 1 diseases-11-00048-f001:**
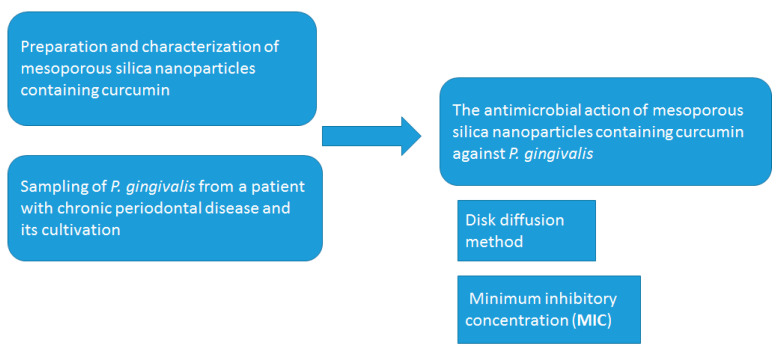
The flow chart of the study process.

**Figure 2 diseases-11-00048-f002:**
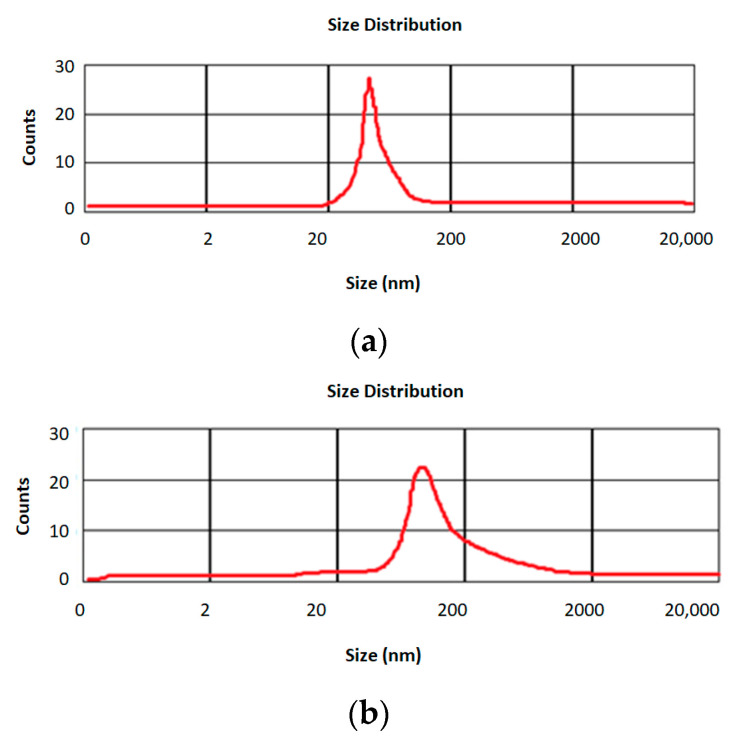
The average particle size for drug-free silica nanoparticles (**a**) and curcumin-loaded silica nanoparticles (**b**).

**Figure 3 diseases-11-00048-f003:**
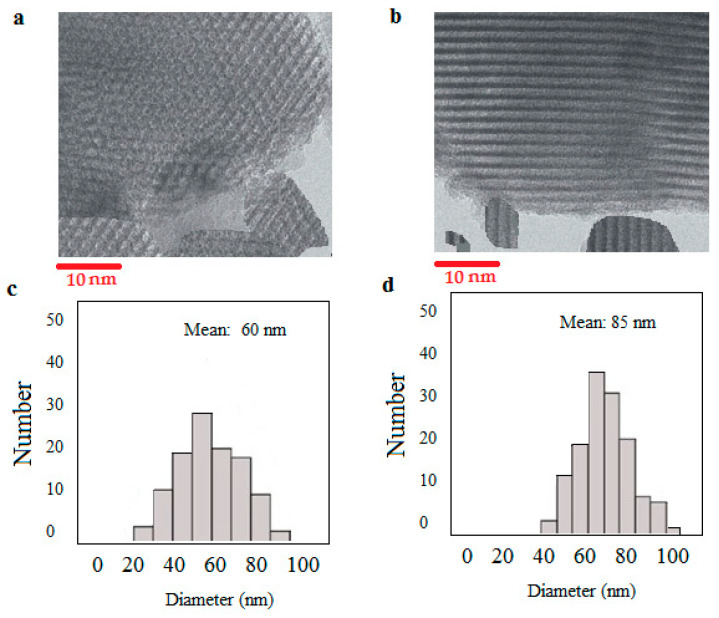
The morphology of drug-free silica nanoparticles (**a**) and the morphology of curcumin-loaded silica nanoparticles (**b**). To present the mesoporous structure of the particles, in the TEM images, only part of a nanoparticle has been illustrated in each case. The size histograms for free silica nanoparticles (**c**) and curcumin-loaded silica (**d**), based on TEM analysis. The histograms illustrate the numbers of particles that were in the field of view of the TEM microscope. The total number of nanoparticles was 269 particles.

**Figure 4 diseases-11-00048-f004:**
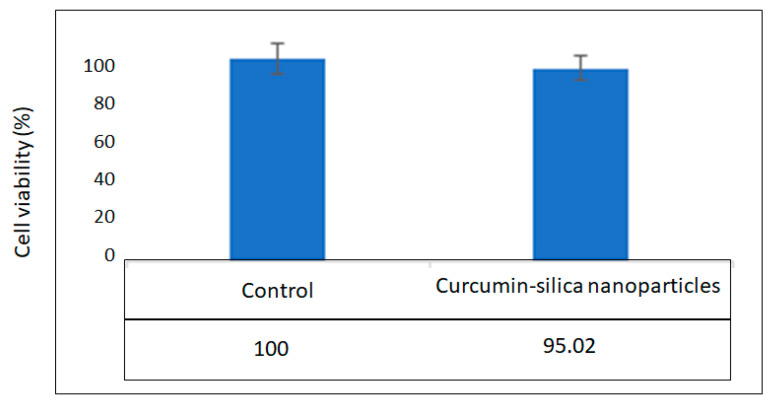
The percentage of cytotoxicity (cell viability) of the prepared nanoparticles on dental pulp stem cells.

**Figure 5 diseases-11-00048-f005:**
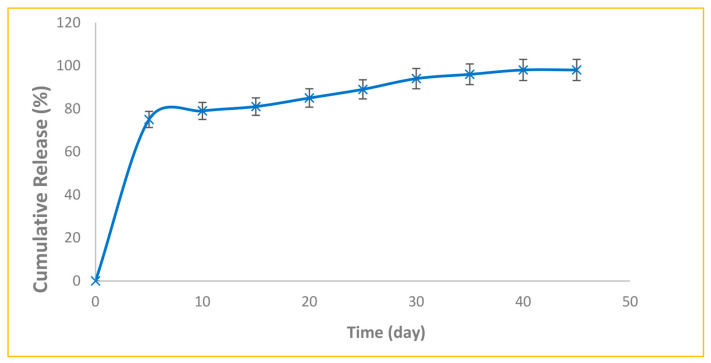
The release profile of curcumin-loaded silica nanoparticles.

**Figure 6 diseases-11-00048-f006:**
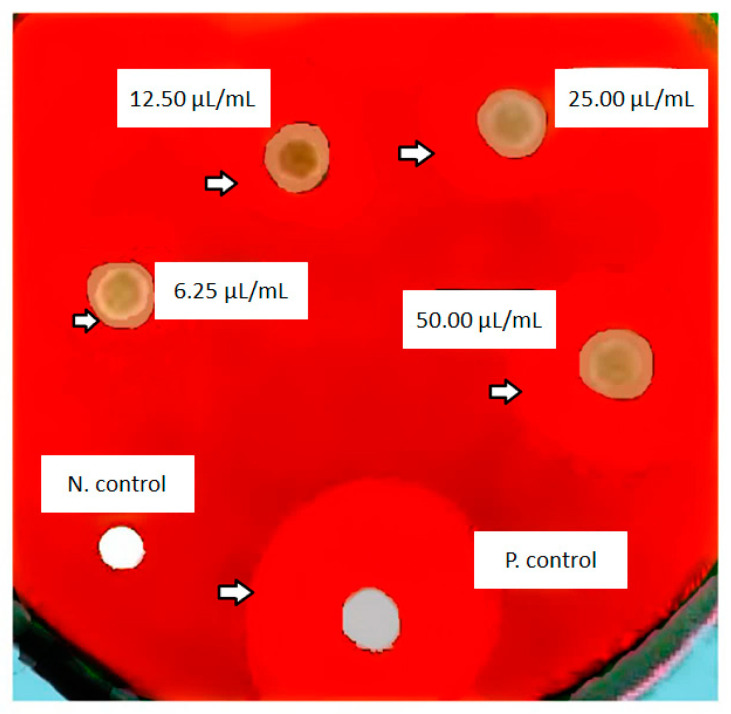
The mean growth inhibition zones of curcumin-loaded silica nanoparticles at different concentrations and in the control groups (negative control (N), positive control (P)).

**Table 1 diseases-11-00048-t001:** The mean growth inhibition zone of curcumin nanoparticle and control groups.

Samples	The Mean Growth Inhibition Zone (mm)
Curcumin-loaded silica nanoparticles (50 µg/mL)	15 ± 1.2
Curcumin-loaded silica nanoparticles (25 µg/mL)	12.23 ± 0.8
Curcumin-loaded silica nanoparticles (12.5 µg/mL)	10.24 ± 1.2
Curcumin-loaded silica nanoparticles (6.25 µg/mL)	7.59 ± 1.4
Metronidazole as the positive control	19.20 ± 1.2
Blank disk (water) as the negative control	0

## Data Availability

The raw data from the reported study are available upon request from the corresponding author.

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
