# Peer review of "Promising Antimicrobial Action of Sustained Released Curcumin-Loaded Silica Nanoparticles against Clinically Isolated Porphyromonas gingivalis"

_diseases, 2023, doi:10.3390/diseases11010048_

Round 1

Reviewer 1 Report (Previous Reviewer 2)

ok

Reviewer 2 Report (Previous Reviewer 4)

NA

This manuscript is a resubmission of an earlier submission. The following is a list of the peer review reports and author responses from that submission.

Round 1

Reviewer 1 Report

The present study determines that local nano curcumin application for periodontal disease and implant-related infections can be considered a promising method in dentistry.  The authors prepare and characterize curcumin-loaded silica nanoparticles and detect its antimicrobial effects on P. gingivalis.

My comments are as follows:

Material and methods:

I am not sure how the bacteria were isolated from the other microorganisms that might be found in the patient's mouth.

This process should be indicated in more detail or the methodology should be referenced.

2.5.4: It would be very didactic and explanatory to add a flow chart of the process.

2.5.5: Is this patient the same as the one in section 2.3?

Style and grammar should be reviewed.

Consider joining sections 7 and 8.

Post-hoc analysis should be performed after ANOVA.

In vitro cytotoxicity should have been evaluated.

Evaluation of proinflammatory cytokine levels of loaded silica nanoparticles should have been evaluated.

Author Response

The present study determines that local nano curcumin application for periodontal disease and implant-related infections can be considered a promising method in dentistry.  The authors prepare and characterize curcumin-loaded silica nanoparticles and detect its antimicrobial effects on P. gingivalis.

My comments are as follows:

Material and methods:

I am not sure how the bacteria were isolated from the other microorganisms that might be found in the patient's mouth. This process should be indicated in more detail or the methodology should be referenced.

It has been done. More details have been added to this section.

2.5.4: It would be very didactic and explanatory to add a flow chart of the process.

It has been done.

2.5.5: Is this patient the same as the one in section 2.3?

Yes. To attain the clinically isolated P. gingivalis, one patient with chronic periodontal disease was selected from the patients referred to the Department of Periodontics, Faculty of Dentistry, Tabriz University of Medical Sciences, Tabriz, Iran.

Style and grammar should be reviewed.

It has been done.

Consider joining sections 7 and 8.

It has been done.

Post-hoc analysis should be performed after ANOVA.

It has been done.

In vitro cytotoxicity should have been evaluated.

It has been done and added.

 Evaluation of proinflammatory cytokine levels of loaded silica nanoparticles should have been evaluated.

Many thanks for your valuable comment. The main purpose of this article was to investigate the antimicrobial effects and properties of the substances synthesized in this study. Therefore, the cost has not considered of doing the molecular tests. Besides, keeping this bacterium is very difficult and its conditions are not available in every place. We do not have it now for further investigation.

Reviewer 2 Report

1. Most of the references (1-20, 25-47, 55-59) are missing the journal title. Unprofessional. 

2. The majority of the manuscript is built as if there is extensive work on the subject worldwide, but referencing predominantly the authors' other published studies (a typical example would be the paragraph with references 25-29). What is on the other hand missing, is an explanation of how this work is different from these other studies from the authors: What is different here from reference 12 or reference 50? Is there an evolution of the previous authors' results or one more publication? 

3. Are there CHX resistance issues? Since CHX is described like a gold standard, what is the rationale of looking for natural antimicrobials in this particular frame?

4. One P. gingivalis isolate, from one patient. Is it enough to conclude on MIC values and accurately compare them to other studies? 

Author Response

Thanks for your valuable comments. 

  1. Most of the references (1-20, 25-47, 55-59) are missing the journal-title. Unprofessional. 

The style of the references was changed by the journal after the submission. Then,  maybe it occurred in that step. We checked and improved.

  1. The majority of the manuscript is built as if there is extensive work on the subject worldwide, but referencing predominantly the authors' other published studies (a typical example would be the paragraph with references 25-29). What is on the other hand missing, is an explanation of how this work is different from these other studies from the authors: What is different here from reference 12 or reference 50? Is there an evolution of the previous authors' results or one more publication?

We corrected the references. This is one more publication. Our previous studies are different from this study. Although they are all in the same direction (antimicrobial effects in the dentistry field). In some cases, we studied curcumin nanocrystals and P. gingivalis, in some cases curcumin nanocrystals and other bacteria, and in some cases curcumin-silica nanoparticles with different bacteria. The sources of bacteria are also different in some cases. In this study, we studied on antimicrobial effects of curcumin-silica nanoparticles on P. gingivalis.

  1. Are there CHX resistance issues? Since CHX is described like a gold standard, what is the rationale for looking for natural antimicrobials in this particular frame?

CHX is the gold standard for dentistry microbial infection prevention and treatment. However, teeth staining is one disadvantage that restricts the long-term use of chlorhexidine. Other side effects include tongue and mucosal surface staining, alterations of taste, desquamation of the mucosa, enlargement of the parotid, and also increased calculus deposition supragingival. Besides, the spread of drug resistance has led to a tendency to natural antimicrobials. We added these new data to the manuscript as well (introduction section).

  1. One gingivalis isolate, from one patient. Is it enough to conclude on MIC values and accurately compare them to other studies? 

We added our MIC results to the revised manuscript. Isolation from one patient is a routine process. Please see the below references:

Pham, Thuy Anh Vu, Thao Thi Phuong Tran, and Ngan Thi My Luong. "Antimicrobial effect of platelet-rich plasma against Porphyromonas gingivalis." International Journal of Dentistry 2019 (2019).‏

McLean, Jeffrey S., et al. "Genome of the pathogen Porphyromonas gingivalis recovered from a biofilm in a hospital sink using a high-throughput single-cell genomics platform." Genome research 23.5 (2013): 867-877.‏

Maleki Dizaj, Solmaz, et al. "Antibacterial Effects of Curcumin Nanocrystals against Porphyromonas gingivalis Isolated from Patients with Implant Failure." Clinics and Practice 12.5 (2022): 809-817.‏

Mendez, Katterinne N., et al. "Variability in genomic and virulent properties of Porphyromonas gingivalis strains isolated from healthy and severe chronic periodontitis individuals." Frontiers in cellular and infection microbiology 9 (2019): 246.‏

Reviewer 3 Report

The authors have made interesting work. They have well designed and executed the experiments; I recommend this article for publication after revising the manuscript based on the following queries.

Authors can perform 16s rRNA sequencing of the clinical isolate to confirm the bacterial strain.

In Section 2.5.4., Please provide the calculation method for the percentage of cumulative drug release of curcumin from silica nanoparticles.

Figure 1d: TEM image of curcumin-loaded silica nanoparticles is not clear. Please provide better images with different magnifications.

Please provide the antimicrobial activity of silica nanoparticles with different concentrations.

Authors should also thoroughly check for any typographical and grammatical errors (for instance check line 170).

Author Response

The authors have made interesting work. They have well designed and executed the experiments; I recommend this article for publication after revising the manuscript based on the following queries.

Thanks for your valuable comments.

Authors can perform 16s rRNA sequencing of the clinical isolate to confirm the bacterial strain.

Thanks for your valuable comments. The main purpose of this article is to investigate the antimicrobial effects and properties of the substances synthesized in this study. Therefore, the cost has not considered of doing the molecular tests. Besides, keeping this bacterium is very difficult and its conditions are not available in every place. We do not have it now for further investigation.

In Section 2.5.4., Please provide the calculation method for the percentage of cumulative drug release of curcumin from silica nanoparticles.

It has been done.

Figure 1d: TEM image of curcumin-loaded silica nanoparticles is not clear. Please provide better images with different magnifications.

It has been done.

 Please provide the antimicrobial activity of silica nanoparticles with different concentrations.

It has been done.

Authors should also thoroughly check for any typographical and grammatical errors (for instance check line 170).

It has been done.

Reviewer 4 Report

Manuscript ID: diseases-2194302

Title: Promising antimicrobial action of sustained released curcumin-loaded silica nanoparticles against clinically isolated Porphyromonas gingivalis

In this article, the authors test the antimicrobial activity of curcumin-loaded silica nanoparticles against clinically isolated Porphyromonas gingivalis. The article is interesting however; some issues must be addressed before it is ready for publication.

11.    The manuscript needs some language corrections.

22.    Please explain why Porphyromonas gingivalis was not identified by the PCR method. For clinical isolates, the molecular identification must be carried out.

33.    Figure 1 must be improved: a) Add legends to the y axis of the hydrodynamic diameter graphs, b) The presented TEM images are not representative of the provided description, for instance in Fig. 1c shape and size of nanoparticles are not distinguished. Please provide new images.

44.    Please provide size histograms for free silica nanoparticles and curcumin-loaded silica, based on TEM analysis. Size of nanoparticles usually differ from that obtained by DLS.

55.    In the manuscript there is only one table, therefore Table 2 should be numbered as Table 1.

66.    In addition to Table 1, it is desirable to provide figures related to the inhibition of bacteria, i.e. antibiograms.

77.    The bacteria are clearly inhibited using the tested concentrations, in a dose dependent manner, however it is desirable to report the minimal inhibitory concentration (MIC).

Author Response

  1. The manuscript needs some language corrections.

Thanks for your valuable comments.

It has been done.

  1. Please explain why Porphyromonas gingivalis was not identified by the PCR method. For clinical isolates, the molecular identification must be carried out.

According to the reviewer’s comment, molecular analysis is useful for identifying bacteria, but standard biochemical and phenotypic tests can be used as a confirmatory method to determine the identity of isolates as mentioned by different reference (Mahon, Connie R., Donald C. Lehman, and George Manuselis. Textbook of diagnostic microbiology-e-book. Elsevier Health Sciences, 2018). As mentioned, in the present study, selective medium for P. gingivalis containing Brucella agar base supplemented with vitamin K1, 5% defibrillated sheep blood, hemin, colistin sulfate, bacitracin, and na-lidixic acid was used as previously described (10.1128/jcm.23.3.441-445.1986). Confirmatory identification of bacterial isolates was done based on standard microbiological tests using a reference and validation of these tests was confirmed using the quality control methods and standard strains. The trypsin reagent test was used to confirm the presence of P. gingivalis in the plates. The gingipain produced by P. gingivalis is one of the trypsin-like enzymes. The aerotol-erance test, biochemical and microbiological assay (such as colony morphology, special potency disks, pigment production, fluorescent under UV light, catalase test, indole, and trypsin-like peptidase activity assay) were used to identify P. gingivalis isolates.

The main purpose of this article is to investigate the antimicrobial effects and properties of the substances synthesized in this study. Therefore, the cost has not considered of doing molecular test. Beside, keeping of this bacterium is very difficult and its conditions are not available in every place. We do not have it now for further investigation.

  1. Figure 1 must be improved: a) Add legends to the y axis of the hydrodynamic diameter graphs, b) The presented TEM images are not representative of the provided description, for instance in Fig. 1c shape and size of nanoparticles are not distinguished. Please provide new images.

It has been done.

  1. Please provide size histograms for free silica nanoparticles and curcumin-loaded silica, based on TEM analysis. Size of nanoparticles usually differ from that obtained by DLS.

It has been done.

  1. In the manuscript there is only one table, therefore Table 2 should be numbered as Table 1.

We corrected. Thanks.

  1. In addition to Table 1, it is desirable to provide figures related to the inhibition of bacteria, i.e. antibiograms.

Thanks for your comment. We added it.

  1. The bacteria are clearly inhibited using the tested concentrations, in a dose dependent manner, however it is desirable to report the minimal inhibitory concentration (MIC).

Thanks for your comment. We added the MIC results.

Round 2

Reviewer 1 Report

The authors have answered most of my questions. In my opinion, the article is an improvement on the first version.

Author Response

The authors have answered most of my questions. In my opinion, the article is an improvement on the first version.

Thanks for your response. We improved the manuscript again. We improved the results presentation as well. Please see.

Reviewer 2 Report

I am satisfied by the response regarding the need for using something further than CHX. 

However, I am not satisfied by the response regarding the added value of the present study, the ways the present study improves the authors' previous publications/ research. Even the authors admit that this is one more publication, when replying to my relevant query. 

The authors fall in the same trap though when advising the reviewer to see that t is a common practice to use a single isolate: aside from the fact that one of the references discusses a biofilm isolate, they again self-reference in order to support what they proclaim as "common practice"

Author Response

I am satisfied by the response regarding the need for using something further than CHX. 

Thanks.

However, I am not satisfied by the response regarding the added value of the present study, the ways the present study improves the authors' previous publications/ research. Even the authors admit that this is one more publication, when replying to my relevant query. 

The authors fall in the same trap though when advising the reviewer to see that t is a common practice to use a single isolate: aside from the fact that one of the references discusses a biofilm isolate, they again self-reference in order to support what they proclaim as "common practice"

Thanks for your comments. You right. Sorry.

Indeed, in our previous study, the P. gingivalis. Bacteria was isolated form patients with implant failure. However, in this study, it was isolated from a periodontitis patient. Also, in the new study, we used silica loaded curcumin nanoparticles and in the previous one we used curcumin nanocrystals.

We add all of our limitation on the limitation section.

Reviewer 3 Report

I recommend the publication of this manuscript after providing a better picture of Figure 5. Either improve the visibility of the zone of inhibitions by Cur-loaded silica nanoparticles or mark a dotted circle for readers' understanding of the diameter of zones.

Author Response

I recommend the publication of this manuscript after providing a better picture of Figure 5. Either improve the visibility of the zone of inhibitions by Cur-loaded silica nanoparticles or mark a dotted circle for readers' understanding of the diameter of zones.

Thanks for your valuable comments. We improved the figure 5.

Reviewer 4 Report

The manuscript has been improved, however there are some details that need to be corrected before publication.

1.     Please improve the presentation of Figures. For instance, Figure 2 can be split in 2 Figures, in order to fit well in the page and make TEM images bigger. The font in Figure 2a and 2b is different. Also, the scale bar in TEM images is missing. TEM images need description in the text, are we looking only part of a nanoparticle in each case?

2.     For the size histograms please specify number of nanoparticles measured.

3.     Please describe in more detail Figures, for instance in Figure 2a and 2b the authors do not mention size obtained, they only mention “The average particle size for drug-free silica nanoparticles is shown in Figure 2a and for curcumin-loaded silica nanoparticles in Figure 2b. The results showed that both types of nanoparticles had nanometric size”.

4.     Figure 3 needs label in y axis. Also needs to be described int the text.

5.     It seems to me that Figure 5 is oversaturated, try to fix brightness/contrast, because inhibition zones are not visible.

Author Response

The manuscript has been improved, however there are some details that need to be corrected before publication.

  1. Please improve the presentation of Figures. For instance, Figure 2 can be split in 2 Figures, in order to fit well in the page and make TEM images bigger. The font in Figure 2a and 2b is different. Also, the scale bar in TEM images is missing. TEM images need description in the text, are we looking only part of a nanoparticle in each case?

Thanks for your valuable comments. We improved the mentioned cases.

  1. For the size histograms please specify number of nanoparticles measured.

Thanks for your valuable comments. We improved the mentioned cases.

  1. Please describe in more detail Figures, for instance in Figure 2a and 2b the authors do not mention size obtained, they only mention “The average particle size for drug-free silica nanoparticles is shown in Figure 2a and for curcumin-loaded silica nanoparticles in Figure 2b. The results showed that both types of nanoparticles had nanometric size”.

Thanks for your valuable comments. We improved the mentioned cases.

  1. Figure 3 needs label in y Also needs to be described int the text.

Thanks for your valuable comments. We improved.

  1. It seems to me that Figure 5 is oversaturated, try to fix brightness/contrast, because inhibition zones are not visible.

Thanks for your valuable comments. We improved.